# Nanomaterials Covered with Cell Membranes for Intracellular Delivery Without Lysosomal Degradation and Innate Immunity Induction

**DOI:** 10.3390/ijms262010244

**Published:** 2025-10-21

**Authors:** Olga Morozova, Ekaterina Obraztsova, Dmitry Klinov

**Affiliations:** 1Federal Research and Clinical Center of Physical-Chemical Medicine Named After Y.M. Lopukhin of the Federal Medical Biological Agency, 1a Malaya Pirogovskaya Street, Moscow 119435, Russia; e.a.obraztsova@gmail.com (E.O.); klinov.dmitry@mail.ru (D.K.); 2Institute of Future Biophysics of Moscow Center of Advanced Studies, 9/7 Institutsky Per., Dolgoprudny 141700, Moscow Region, Russia; 3National Research Center of Epidemiology and Microbiology of N.F. Gamaleya of the Russian Ministry of Health, 16 Gamaleya Street, Moscow 123098, Russia

**Keywords:** extracellular vesicles, low-speed centrifugation, scanning transmission electron microscopy (STEM), dynamic light scattering (DLS), confocal fluorescent microscopy, lysosome staining, interferon (IFN) gene expression

## Abstract

Cellular uptake of nanomaterials is based on endocytosis with their endosomal–lysosomal entrapment resulting in enzymatic hydrolysis. Besides biodegradation, the antigen presentation induces innate and adaptive immunity. Our goal was isolation of extracellular particles to study their structures, penetration into cells, stability, intracellular distribution, and interferon (IFN) production. Extracellular nanomaterials were isolated from conditioned culture media of human embryonic and cancer cells by two-stage differential centrifugation. Cellular uptake of Cy5-labeled particles was evaluated using spectrofluorimetry and confocal fluorescent microscopy. IFN gene expression was analyzed by reverse transcription with real-time PCR and ELISA. Vesicles of 10–200 nm were isolated by centrifugation at 20,800× *g* at +4 °C for 30 min. The fluorescent vesicles were gradually accumulated inside cells for seven days. Intracellular distribution patterns of the Cy5-labeled vesicles differed from lysosomes stained with LysoRed tracker. IFNs α, β and γ were not detected after treatment with the vesicles. IFN λ was found in cells in the presence of allogenic but not autologous particles. The gradual cellular uptake occurred without significant differences between autologous and heterologous vesicles. Different localization of the extracellular vesicles (EV) and lysosomes along with weak innate immune response (if any) suggested membrane fusion.

## 1. Introduction

According to the International Society for Extracellular Vesicles guidelines and the third iteration of the ‘Minimum Information for Studies of Extracellular Vesicles’ (MISEV) MISEV2023 [1], the term of the extracellular particles (EPs) combines both extracellular vesicles (EV) and non-vesicular extracellular particles without lipid bilayers. EV are defined as particles that are released from cells, are delimited by a lipid bilayer, and cannot replicate on their own [1]. EV encompasses all types of membrane-bound vesicles within the broad size range from 10 to 1000 nm that are released from cells. General structural traits of EV include their envelopes consisting of cellular membranes and the encapsulation of various biomolecules into their internal space. EV are secreted by cells to function in intercellular communications, cell proliferation, differentiation, migration, apoptosis, immunomodulation, and the ability to cross physiological barriers (plasma membrane, blood–brain barrier, and gastrointestinal tract) [2]. EV may transfer both materials and biological signals including nucleic acids, proteins, cytokines, and others [3] between cells via body fluids, among different species, and even various kingdoms of life ([4] and references therein). Mammalian-derived vesicles possess bioactivity and precise regulatory capabilities for therapy, while plant- and insect-derived counterparts may fill gaps in conventional treatment by means of natural intracellular delivery, antioxidant molecules, and antimicrobial peptides [2]. Plant-derived EV can alter the physiological activities of animal cells [5]. EV are found in most biological fluids, such as blood, urine, saliva, cerebrospinal fluid, pleural effusion, ascites, amniotic fluid, semen, bile, milk, and even gastric juice [2]. Thus, blood contains a large number of extracellular lipoprotein particles, urine contains abundant Tamm-Horsfall proteins, and bronchoalveolar lavage fluid contains surfactants. Brain-derived EV after crossing the blood–brain barrier can be found in the blood and used as biomarkers of central nervous system diseases and neuroinflammation without a brain tissue biopsy or a lumbar puncture [6]. The vesicles can originate and release from various intracellular compartments, including surface plasma, endosomal and non-endosomal membranes [7]. Therefore, their sizes, shapes and biochemical compositions are heterogeneous. The tetraspanins CD9, CD63, and CD81 were earlier suggested to be major EV components. Therefore, fluorescent antibodies against CD9, CD63, and CD81 were previously used for immunochemical identification ([8] and references therein). But recently, CD63 was shown to be predominantly intracellular. In contrast, CD9 and CD81 are localized at the plasma membrane. Absence of these marker proteins had little impact on the EV composition as analyzed by quantitative mass spectrometry [8].

EV can be classified based on their sizes as small (20–200 nm), large EV (200 nm–10 μm) and lipid droplets (40 nm–100 μm). In turn, the small EV include supermeres (>25 nm), exomeres (>50 nm), exosomes (40–200 nm), defensomes (~80 nm), and partly microvesicles (100 nm–1 μm), mediate intra- and intercellular communications, and are involved in numerous physiological and pathological processes. Because of overlapping size ranges of different types of small EV, their complete and exact differentiation and identification on the basis of only size measurements are hardly possible. Additional multiplex immunochemical analysis of transmembrane proteins is required. Exosomes are formed by the inward budding of endosomal membranes resulting in multivesicular bodies and fuse with the plasma membranes to release them into the extracellular space [7].

EV can be isolated and purified by a variety of methods. The main EV extraction and purification methods include differential ultra-high-speed centrifugation, sucrose gradient density centrifugation, immunoaffinity capture, ultrafiltration, size exclusion chromatography, polyethylene glycol (PEG) co-precipitation, and others [2,9]. The yield and purity of the EV depend on both their source and the processing methods. A primary challenge in isolating EV is removing nanoscale contaminants. Ultracentrifugation remains the most widely employed technique for isolating small EV. Other isolation techniques, such as size exclusion chromatography and PEG co-precipitation, are used in clinical samples for their convenience. Ultrafiltration is cost-effective and fast.

Natural EV can be used for diagnostics as important biomarkers. Due to liquid biopsy sampling accessibility, EV may become potential biomarkers [3] but their diagnostic and therapeutic exploitation in basic research and clinical implementation is currently restricted by the labor-intensive, inefficient, and time-consuming isolation procedures. For therapy, the EV must be loaded with drugs. Bioinspired vesicles represent a class of artificial nanocarriers with structural and functional characteristics of natural vesicles including compositional architecture, dynamic behaviors, and signaling mechanisms of cellular/organelle membranes. Compared to natural EV, these engineered systems exhibit structural stability, drug-loading efficiency, and functional programmability, while being capable of overcoming inherent challenges of natural vesicles such as source-dependent heterogeneity and scalability limitations in manufacturing. The bioinspired vesicles show significant therapeutic effects [2].

Lipids, as the main components of EV, play a role in maintaining structural stability and participating in biological functions of vesicles. Membrane structures of EV are similar to the membrane structure of cell membranes, and contain a variety of lipid components, such as triglycerides, diacylglycerol, phosphatidylcholine, and phosphatidylinositol. Diacylglycerol and phosphatidic acid were previously reported to have a significant effect on membrane fusion and vesicle uptake [9]. Transmembrane proteins and glycoproteins of EV are classified into functional, structural, and secreted proteins according to their functions. Functional proteins are involved in many biological processes such as cell expansion, protein folding, and stress (in particular, heat shock). Wheat-derived EV contain heat shock proteins and other chaperones that prevent stress and mediate protein folding [9].

Our research was aimed at the isolation of extracellular particles from conditioned culture media after growth of adherent human embryonic and cancer cells, analysis of their sizes, shapes, chemical composition, stability, cellular uptake, subcellular localization, as well as the innate immune response.

## 2. Results

### 2.1. Isolation of EV from Culture Media

Extracellular particles were isolated from the conditioned culture media of the adherent human embryonic (both embryonic fibroblasts and lung cells) and cancer (HeLa, HEp-2 and L41) cells after their growth until confluent monolayers by centrifugation at 20,800× *g* at +4 °C for 30 min without additional procedures of ultracentrifugation and ultrafiltration. Scanning transmission electron microscopy (STEM) and dynamic light scattering (DLS) showed their diameter range of 10–200 nm with internal coreless space covered with cell membrane of ~7–10 nm (Figure 1).

The chemical composition of the vesicles was determined by means of spectrophotometry. The protein concentrations varied in the range 0.28 ÷ 0.32 mg/mL for the embryonic cells and 0.41 ÷ 0.99 mg/mL for the cancer cells as determined by UV absorption at 280 nm. Fluorescent labeling of the particles with the reactive Cy5 isothiocyanate derivatives (Figure 1) also confirmed the presence of proteins. Phospholipids forming a complex with ammonium ferrothiocyanate were defined colorimetrically [10,11]. On the basis of the calibration curve with known concentrations of pure isolated phospholipids, the corresponding concentrations in the EV samples were from 25 to 125 nmole/mL.

The vesicles were stable during storage for up to five months of observations in PBS at +4 °C and in the fluorescent labeling reaction at pH 9.3 (Figure 1).

However, incubation in deionized water, freeze–thaw cycles and sonication disrupted their membrane envelopes with leakage of their internal content. Additional washes with deionized water or PBS with subsequent centrifugation at 20,800× *g* helped to remove the internal ingredients of the broken EV. The simple and fast procedures are convenient for the EV envelopes purification for the subsequent loading with drugs and composite membrane-covered nanomaterials construction by means of the “thin film hydration” approach.

### 2.2. Thin Film Hydration

Bioinspired nanoparticles consisting of internal core protein or gold nanoparticles and surface EV-derived membrane shells were constructed by the thin film hydration method (Figure 2).

### 2.3. Cellular Uptake of the Fluorescent Particles

Transmembrane proteins were labeled with the fluorescent cyanine Cy5 dye and the resulting fluorescent EV were purified by additional centrifugations at 20,800× *g* for 30 min. Despite their small sizes of less than 200 nm, the fluorescent EV were partly visible using fluorescent microscope with magnifications 600× (Figure 1) or more.

Autologous and heterologous EV isolated from the conditioned culture media of human cells of different origin and the cyanine Cy5 dye were not toxic for all cell lines studied probably because of low concentrations of proteins and lipids in the isolated nanomaterials. The fluorescent vesicles gradually accumulated inside the adherent human cells over seven days of observation as shown by spectrofluorimetry (Figure 3) and confocal fluorescent microscopy (Figure 4 and Figure 5). Noteworthy that a part of the fluorescent vesicles could be found outside the embryonic cells in 3–7 days posttreatment whereas the same particles were mainly inside the human cancer cells HeLa and HEp-2 in 3 days after EV addition, probably due to the enhanced penetration and retention (EPR) effect, which is a feature of cancer cells ([4] and references therein).

Distribution patterns of the fluorescent EV in the embryonic (Figure 4) and cancer (Figure 5) cells were analyzed using confocal fluorescent microscopy. All intracellular EV were outside cellular nuclei stained with the DNA intercalating dye Hoechst 33342 and their localization patterns did not coincide with lysosomes stained with LysoRed lysosomal tracker (Figure 4 and Figure 5).

### 2.4. Induction of Innate Immunity

Due to low protein (~0.3 mg/mL) and phospholipid (~10 nmole/mL) concentrations (in wells of culture plates after addition of one-tenth of the well volume)along with flexible conformations of lipids, the innate immune response was very weak (if any). The prolonged IFN response for several days without evident peak (Table 1) could result from the gradual penetration of EV into human cells (Figure 3) and trained innate immunity. Part of available data is shown in Table 1. After addition of autologous EV to their host cells, IFN gene expression was not detected by RT^2^-PCR and ELISA. Treatment of the human embryonic fibroblasts with EV isolated from cancer HEp-2 cells caused IFN λ RNA transcription and production of the encoded protein (Table 1) whereas IFN α/β of type I and IFN γ of type II were not detected.

## 3. Discussion

Current limitations for clinical translation of biological fluid-derived vesicles include isolation technical constraints due to their inherent heterogeneity including dimensional variability (10–1000 nm), transmembrane protein compositional differences, and cargo diversity resulting in various densities. Despite ultracentrifugation (including differential centrifugation and density gradient ultracentrifugation) remaining the most widely used EV isolation “gold standard”, the technique yields only moderately pure samples and is time-consuming. The relative purification can be achieved by step-by-step differential centrifugations including a low-speed centrifugation (300–400× *g*) to remove cellular debris, an additional centrifugation (2000× *g*) to eliminate membrane debris and larger vesicles, and an ultra-fast centrifugation to harvest the small EV. However, it is difficult to avoid EV loss at each stage. Moreover, the conventional ultracentrifugation causes structural damages of vesicle membranes and insufficient recovery rates of 5–25% as quantified by nanoparticle tracking analysis [2]. Blood serum protein aggregates and lipoprotein particles can be co-precipitated together with the EV. In addition, the ultracentrifuges are expensive and, therefore, are not available in a number of clinical centers. To overcome the main obstacles of ultracentrifugation such as low separation yields, potential safety concerns, and unsatisfactory production, we isolated the small EV by using the fast and simple centrifugation at 20,800× *g* for 30 min at +4 °C instead of long repeated ultracentrifugations at 100,000× *g*. The EV density of 1.13–1.19 g/mL permitted them to be precipitated without structural damages (Figure 1) using common bench centrifuges.

The size range of 10–200 nm estimated by STEM and DLS measurements (Figure 1) proved that the isolated vesicles belonged to the small EV. Classification of the small EV types including exosomes (40–200 nm), defensomes (~80 nm), exomeres (>50 nm), and supermeres (>25 nm) based on their sizes alone is impossible. Additional multiplex analysis of the transmembrane proteins by immunochemical methods is required. Microvesicles (100 nm–1 μm), oncosomes (100 nm–10 μm) and migrasomes (up to 3 μm) may be excluded from consideration because of their diameter ranges exceeding the experimentally observed values (Figure 1). Oncosomes are known to be specific for cancer cells only but EV of similar structures and sizes were isolated from the conditioned culture media after growth of both embryonic and cancer cells. Exomers do not have a lipid bilayer membrane, which did not correspond to our STEM images (Figure 1). Migrasomes look like a pomegranate with many small bubbles. Therefore, the main types of isolated EV were exosomes, defensomes and supermers.

The EV stability during storage in cold PBS under refrigeration up to five months of our observations offers an evident advantage both for diagnostic monitoring and for the nanotechnological loading with therapeutic drugs for personalized nanomedicine. However, incubation in deionized water, freeze–thaw cycles, and sonication disrupted the EV membrane integrity, leading to content leakage. The simple and fast procedures facilitate EV envelope purification, enabling subsequent drug loading and the construction of composite membrane-coated nanomaterials via “the thin-film hydration” approach. Due to high lipid solubility, the EV increases hydrophobic drug absorption and drug membrane permeability [9]. For targeted delivery, the bioinspired vesicles loaded with drugs can be modified through genetic engineering and chemical modifications [2].

Despite the targeted delivery of EV to organs with their higher concentrations in donor organs than in other organs, the molecular mechanisms of the enhanced therapeutic effect [9] remain unclear. Our data did not reveal essential advantages in cellular uptake of the fluorescent autologous compared to heterologous EV (Figure 3, Figure 4 and Figure 5).

Cellular uptake of nanomaterials is mainly based on endocytosis with invaginations of the plasma membrane to create vesicles with their subsequent fusion with endosomes and later with lysosomes [12], where lysosomal enzymes hydrolyze foreign proteins, nucleic acids, carbohydrates, and lipids. An alternative way, membrane fusion, may avoid endosomal–lysosomal entrapment and enzymatic hydrolysis. Membrane fusion takes place when oocytes are fertilized by sperm cells. The formation of hybrid cells, in particular, hybridomas secreting monoclonal antibodies, also involves membrane fusion. Enveloped viruses require membrane fusion to enter host cells [13]. Lipid bilayer fusion requires catalysis to overcome a high kinetic barrier. Viral surface fusion glycoproteins are responsible for the catalytic function [13]. The membrane fusion sometimes leads to formation of a multinuclear syncytium. Vesicular intra- and intercellular transport is also based on membrane fusion. The mechanism can be used for intracellular delivery of stable nanomedical drugs [4] without their lysosomal biodegradation (Figure 4 and Figure 5).

Besides biodegradation of the internalized nanoparticles, the foreign antigen presentation induces innate and adaptive immunity with further elimination of both nanostructures and host cells. The vesicles display minimal immunogenicity (Table 1). The most potent immunogens are known to be proteins and polysaccharides whereas lipids and nucleic acids are weak immunogens (if any) due to their flexible conformations in solutions. Among three types of human interferons (IFN), only IFN λ (IL29) was detected by RT^2^-PCR and ELISA after the incubation with heterologous EV. It is hardly possible to estimate polarization indexes for so weak innate immune response (Table 1). IFN λ, known for its antiviral activity at mucosal surfaces, can influence the composition and function of EV [14], and, in turn, these EV can impact the immune landscape (Table 1). EV possess an innate capacity to evade recognition as shown in Table 1, transport and transfer functional components to target cells, with subsequent removal by the immune system, where the immunological activities of EV impact immunoregulation including modulation of antigen presentation and cross-dressing, immune activation, immune suppression, and immune surveillance, impacting the tumor immune microenvironment [15]. EV from immune, tumor, and stromal cells and even bacteria and parasites mediate the communication of various immune cell types to dynamically regulate host immune response [15]. Bacterial EV can modulate the host immune system and vice versa; cellular EV can participate in antimicrobial immunity [2]. Heterologous EV of different origins are capable of transferring nanomaterials in foreign cells due to weak immune rejection. Despite evident advantages of plant-derived vesicles such as their relative biosafety compared to animal-derived products, cost effective, fast, and easy isolation as well as the absence of ethical issues [5], the transmembrane proteins of plant membranes might induce an undesirable innate and adaptive immune response with removal of foreign nanomaterials and whole cells and possible development of allergic, autoimmune, and inflammatory diseases.

## 4. Further Development

Natural and bioinspired loaded vesicles are suggested to be a promising alternative strategy for intracellular delivery. Cellular membranes of the vesicles protect natural or artificially loaded internal materials from degradation without immune rejection, tumorigenicity, teratogenicity, and ethical problems. Biological vesicles are suited for long-term storage due to their high stability in refrigerators. Transmembrane proteins of the EV may mediate targeted delivery, stimulate cell migration, proliferation, and differentiation, accelerate the formation of blood vessels and extracellular matrix reconstruction [2]. However, leakage of inner substances through membranes cannot be excluded from consideration. Core–shell structures with internally very stable protein nanoparticles (including other bioactive additives) and membrane envelopes that help to escape endocytosis, lysosomal biodegradation and immune rejection seem to be promising for further innovations and clinical implementation.

## 5. Materials and Methods

### 5.1. Cell Cultures

Human embryonic fibroblasts and lung cells from the Russian State Tissue Culture Collection (National Research Center of Epidemiology and Microbiology, Moscow, Russia) were grown in Minimum Essential Medium Eagle (MEM) with 7% fetal calf serum (FCS) (HyClone, Thermo Fisher Scientific, Waltham, MA, USA) supplemented with L-glutamine, 50 units/mL penicillin, and 50 μg/mL streptomycin until 80% confluent monolayers formation. Human carcinoma of the cervix cell line HeLa, human larynx carcinoma HEp-2 and oral epithelial carcinoma L41 cells were obtained from the Russian State Tissue Culture Collection (National Research Center of Epidemiology and Microbiology, Moscow, Russia) and grown in culture medium 199 supplemented with 7% FCS in the presence of the same antibiotics and L-glutamine until sub confluent monolayers.

### 5.2. EV Isolation

EV were isolated from the conditioned culture media of the human embryonic and cancer cell lines by means of two-stage differential centrifugations at 9000× *g* for 15 min to remove membrane debris and 20,800× *g* for 30 min at +4 °C to precipitate EV [4,11]. To remove possible unspecific contamination with extracellular nanoparticles, the EV pellets were additionally washed with cold PBS 3 times with subsequent sedimentation at 20,800× *g* at +4 °C during 30 min each.

### 5.3. Scanning Transmission Electron Microscopy (STEM)

Samples of EV in PBS were deposited onto 400 mesh copper grids with formvar, stabilized with a carbon support film by floating on a drop of a sample for 3–5 min at room temperature. Then, negative staining with 2% uranyl acetate was performed for 1 min, and the remaining liquid was removed via filter paper. EV were visualized using a Zeiss Merlin microscope equipped with GEMINI II Electron Optics (Zeiss, Oberkochen, Germany) with an accelerating voltage of 20 kV and a probe current range of 80–200 pA.

### 5.4. Dynamic Light Scattering (DLS)

EV sizes were determined by DLS using NANO-flex 180° (Microtrac, Montgomeryville, PA, USA). Refraction index (n) of proteins was 1.4 and n (water) = 1.3.

### 5.5. Ultraviolet (UV) Spectroscopy

UV absorption spectra of EVs and membrane fractions were measured using NanoDrop 2000c UV-Vis spectrophotometer (Thermo Fisher Scientific, Waltham, MA, USA). Phospholipids forming a complex with ammonium ferrothiocyanate were defined colorimetrically [10].

### 5.6. Fluorescent Labeling and Quantitation

Membrane fractions and EV were preliminary labeled with the fluorescent cyanine dye Cy5 (Sigma-Aldrich, Saint Louis, MO, USA) in 0.1 M Na_2_CO_3_ solution pH 9.3 for 1 h at room temperature and purified from the dye by Sephadex G25 chromatography. Cytopainter Lysosomal Staining kit–Red (ab112137, Abcam, Cambridge, MA, USA) was used to visualize cellular lysosomes according to the manufacturer’s instructions.

Fluorescence emission was measured using spectrofluorometer “Fluoromax+” (Horiba Scientific, Kyoto, Japan).

### 5.7. Thin Film Hydration

The isolated EV in isotonic PBS were diluted and vigorously mixed in 10 volumes of deionized water to destruct the vesicles under hypo-osmotic conditions. Then, the mixtures of the broken EV were dried in thin films under reduced pressure at 37–50 °C. Subsequent addition of colloid solutions of protein or gold nanoparticles fabricated as earlier described ([4] and references therein) resulted in composite core–shell nanomaterials [4,11].

### 5.8. Confocal Microscopy

Before confocal microscopy culture media were removed, attached cells of adherent human embryonic fibroblasts, embryonic lung cells and cancer cell lines were washed twice with cold PBS and fixed with 4% formaldehyde solution in PBS for 10 min at room temperature. Then, the cell nuclei were additionally strained with Hoechst 33342 (Abcam, USA).

Microscopic observations were performed using a confocal laser scanning microscope OLYMPUS FV3000 (OLYMPUS, Tokyo, Japan) equipped with a 60×/1.42 oil-immersion objective lens. The different channels were recorded as follows: excitation at 405 nm, emission wavelength range 410–510 nm corresponding to Hoechst 33342; excitation at 561 nm, emission at 570–670 nm for Alexa-Fluor 568 to detect LysoRed indicator (Abcam, USA); excitation at 640 nm, emission range 650–750 nm for Cy5 detection. The power of all lasers was set in a range 3–5% of the nominal power; the gain and the offset were set to the same value. For standard image processing and presentation, ImageJ 1.54p (https://imagej.net) (accessed during period 2023–2025) software was used.

### 5.9. Reverse Transcription with Real Time PCR (RT^2^-PCR)

Total nucleic acids were isolated from 100 μL of control intact and experimental cells using “Proba-NK” kit (“DNA-technology”, Moscow, Russia). Then, the reverse transcription with random hexamer primer was performed using “Reverta-L” kit (AmpliSens, Moscow, Russia). RT^2^-PCR to detect mRNA of human interferons IFN α, β, γ, λ with specific primer pairs and fluorescent hydrolysis probes (Table 2) was performed as previously described [16]. Quantitation of genome-equivalents was based on Lukyanov–Matz’s equation and a calibration curve with the standards.

### 5.10. Enzyme-Linked Immunosorbent Assay (ELISA)

IFN α and IFN γ were detected using corresponding ELISA kits produced by «Vector-Best», https://vector-best.ru, Novosibirsk, Russia. Human IFN β was evaluated in ELISA using “Human IFN β (Interferon beta)” kit (FineTest www.fn-test.com, Wuhan Fine Biotech. Co., Ltd., Wuhan, China) (accessed on 11–15 August 2025). For IFN λ the ELISA kit “Interleukin 29 (IL29)” (Cloud-Clone Corp. www.cloud-clone.com, Wuhan, China) (accessed on 11–15 August 2025), was used.

## 6. Conclusions

EV were isolated by means of two-stage low-speed differential centrifugation of the conditioned culture media after growth of adherent human embryonic and cancer cells using bench microcentrifuge without additional high-speed ultracentrifugations. STEM and DLS proved that the isolated EV belong to the small EV with a size range 10–200 nm. The EV were stable during storage in PBS in a refrigerator for up to five months of observation. However, incubation in deionized water, freeze–thaw, and sonication disrupted EV membranes. The envelopes could be dried in thin layers under reduced pressure and used for loading by the “thin film hydration” approach. The EV were not toxic for human cell lines studied and gradually accumulated inside attached cells in monolayers for seven days without peak values. Intracellular distribution of the fluorescent EV did not coincide with lysosomes as shown by confocal fluorescent microscopy. The EV escape from lysosomal entrapment and biodegradation did not result in antigen presentation and innate immune response induction. The interferon IFN λ gene expression was found after addition of heterologous EV from foreign cells only.

## Figures and Tables

**Figure 1 ijms-26-10244-f001:**
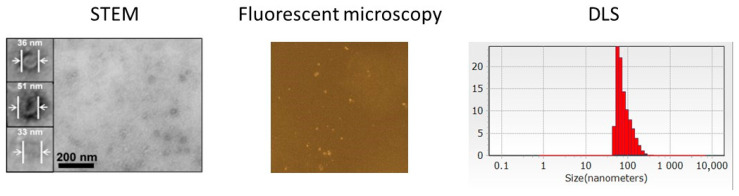
Detection of the natural extracellular vesicles (EV) isolated from the conditioned culture medium after growth of the human embryonic fibroblasts by means of scanning transmission electron microscopy (STEM), fluorescent microscopy of Cy5-labeled natural EVs, and dynamic light scattering (DLS). Magnification 600×.

**Figure 2 ijms-26-10244-f002:**
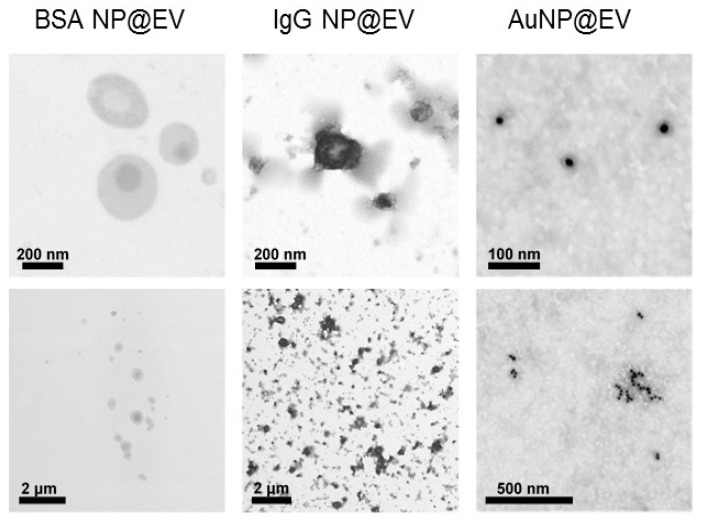
STEM images of bioinspired vesicles consisting of protein (bovine serum albumin and human immunoglobulin IgG) and gold nanoparticles covered with cellular membranes from extracellular vesicles.

**Figure 3 ijms-26-10244-f003:**
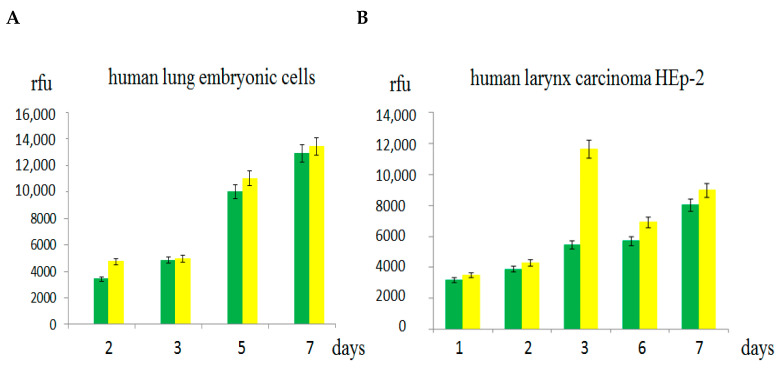
Dynamics of intracellular accumulation of the fluorescent Cy5-labeled extracellular vesicles isolated from the human embryonic lung cells (green columns) and the human larynx carcinoma HEp-2 cells (yellow columns) in the human embryonic lung cells (**A**) and cancer HEp-2 cells (**B**). Average values of Cy5 fluorescence emission from three independent experiments with standard deviations are shown on the axis Y as relative fluorescence units (rfu).

**Figure 4 ijms-26-10244-f004:**
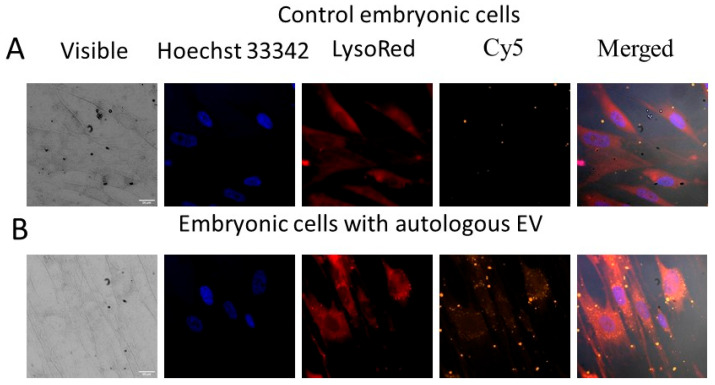
Confocal fluorescent microscopy images of the human embryonic fibroblasts stained with LysoRed lysosomal tracker in three days posttreatment with the Cy5-labeled extracellular vesicles. Size scale is 20 μm. Part (**A**) shows control embryonic fibroblasts; part (**B**)—the embryonic fibroblast cells after addition of Cy5-labelled EV. Nuclei of cells were stained with the blue fluorescent dye Hoechst 33342. Lysosomes were stained with the red fluorescent dye LysoRed. Magnification 600×.

**Figure 5 ijms-26-10244-f005:**
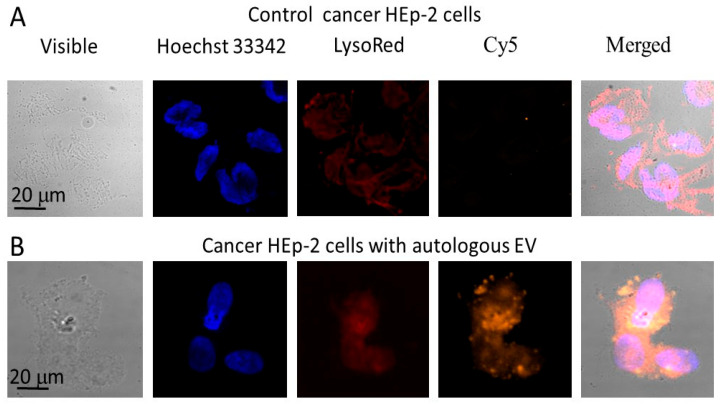
Confocal fluorescent microscopy images of the human larynx carcinoma HEp-2 cells stained with LysoRed lysosomal tracker in three days posttreatment with the Cy5-labeled extracellular vesicles. Part (**A**) shows control cancer HEp-2 cells; part (**B**)—the cancer HEp-2 cells after addition of Cy5-labelled EV. Nuclei of cells were stained with the blue fluorescent dye Hoechst 33342. Lysosomes were stained with the red fluorescent dye LysoRed. Magnification 600×.

**Table 1 ijms-26-10244-t001:** Detection of IFN λ RNA and protein in the human embryonic fibroblasts (HEF) and culture medium, respectively, after treatment with the EV isolated from the culture medium of cancer HEp-2 cells.

Days After Treatment of HEF with the Heterologous EV and Cellular Membranes	IFN λ RNA Ct (Genome-Equivalents per Cell)	IFN λ (pg/mL)
EV 1 day	21.4 (119)	62.29
EV 3 days	20.2 (273)	67.36
Cell membranes 1 day	21.1 (147)	68.41
Cell membranes 3 days	21.8 (90)	71.2

**Table 2 ijms-26-10244-t002:** Structures of primers and fluorescent hydrolysis probes (5′-3′ end) for RT^2^-PCR.

Names of Primers and Fluorescent Probes	Nucleotide Sequences (5′-3′ End)
IFN α—F	AAATACTTCCAAAGAATCAC
IFN α—R	AAGAGAGGGATCTCATG
IFN α—P	FAM-CTGACA ACCTCCCAGGCACAAG-BHQ1
IFN β—F	GATTCTGCATTACCTGAAG
IFN β—R	AGGTAACCTGTAAGTCTG
IFN β—P	Cy3-GCCTGGACCATAGTCAGAGTGG-BHQ2
IFN γ—F	GGAGACCATCAAGGAAGA
IFN γ—R	GACTTGAATGTCCAACGCAAAGC
IFN γ—P	R6G-GACTTGAATGTCCAACGCAAAGC-BHQ2
IFN λ—F	CTGCAGGTGAGGGAGCGC
IFN λ—R	CAGGGTGTGAAGGGGCTG
IFN λ—P	Cy5-GAGGCTGAGCTGGCCCTGACGC-BHQ2

## Data Availability

The original contributions presented in this study are included in the article. Further inquiries can be directed to the corresponding author(s).

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
