# Peer review of "Nanomaterials Covered with Cell Membranes for Intracellular Delivery Without Lysosomal Degradation and Innate Immunity Induction"

_ijms, 2025, doi:10.3390/ijms262010244_

Round 1

Reviewer 1 Report

Comments and Suggestions for Authors

The manuscript investigates the mechanisms underlying the cellular uptake of extracellular vesicles (EVs), their subsequent intracellular trafficking, and the ability of these vesicles to bypass endo-lysosomal degradation and activate innate immune responses. This topic holds significant relevance in the fields of nanomedicine and targeted drug delivery, especially for therapeutic EV applications. However, several methodological and interpretative aspects warrant further elucidation, and certain conclusions drawn lack robust experimental support. Additionally, enhancements in clarity, structure, and overall scientific rigor are necessary to strengthen the manuscript’s impact.

Revise keywords to use more accessible language.

The title does not accurately reflect the primary findings of the text.

The abstract requires refinement; the main objective is unclear, and the technical language needs simplification.

While the introduction is engaging, it lacks clarity. The authors should aim for conciseness by reducing the length of some paragraphs. Additionally, the research problem and hypothesis need to be articulated more clearly. Please improve the readability of this section.

Regarding references, please clarify the phrase “[3 and references therein].”

The drying conditions for the EV film appear to be critical. Can the authors confirm that the structures remain consistent with the initial evaluation? What potential impacts might there be on their physicochemical and biological properties?

It would be beneficial for the authors to include a figure that outlines their experimental design to illustrate the steps of the study more effectively.

Were any steps taken to remove protein aggregates or non-EV contaminants from the centrifuged samples?

How was the gradual accumulation of EVs quantified? Were fluorescence intensity values normalized?

Were appropriate positive controls (e.g., LPS, Poly I:C) used to validate the interferon gene expression system?

No statistical analysis was provided. It is essential to address this to better demonstrate the effects of size and the significance of the findings.

Please revise figure captions to correct typographical errors and address any misuse of abbreviations.

For clarity, rather than referring to “lower/upper panel,” identify images as A, B, C.

Figures 4 and 5 currently have suboptimal resolution; please revise accordingly.

The methodology for evaluating innate immunity activation was inadequately described, leading to confusion in the results. This section should be revisited.

The discussion lacks depth and largely repeats the findings without adequately linking them to relevant literature. A more comprehensive analysis is needed.

The assertion that EVs evade immune detection warrants a more cautious interpretation until comprehensive immune assays, such as cytokine profiling and assessments of antigen presentation corroborate it.

Authors are invited to include a brief discussion regarding potential limitations of their findings in light of proper data interpretation.

Comments on the Quality of English Language

The manuscript contains several grammatical errors and fragmented phrasing; a thorough language review is necessary.

Author Response

The manuscript investigates the mechanisms underlying the cellular uptake of extracellular vesicles (EVs), their subsequent intracellular trafficking, and the ability of these vesicles to bypass endo-lysosomal degradation and activate innate immune responses. This topic holds significant relevance in the fields of nanomedicine and targeted drug delivery, especially for therapeutic EV applications. However, several methodological and interpretative aspects warrant further elucidation, and certain conclusions drawn lack robust experimental support. Additionally, enhancements in clarity, structure, and overall scientific rigor are necessary to strengthen the manuscript’s impact.

Answer:

Thank you for your time and attention. Your careful detailed analysis helped us to improve our revised manuscript.

Revise keywords to use more accessible language.

Answer:

Keywords have been revised accordingly.

Keywords: extracellular vesicles (EV); low-speed centrifugation; scanning transmission electron microscopy (STEM); dynamic light scattering (DLS); confocal fluorescent microscopy; lysosome staining; interferon (IFN) gene expression.

The title does not accurately reflect the primary findings of the text.

Answer:

The title has been changed to the following.

“Extracellular vesicles for intracellular delivery without lysosomal degradation and innate immunity induction”.

The abstract requires refinement; the main objective is unclear, and the technical language needs simplification.

Answer:

The Abstract was corrected according to IJMS instructions and opinions of both Editor and Reviewers. Scientific terms are common, customary and widely used. The number of words in the Abstract is limited (less than 200 words) and does not permit to describe all experimental details along with main results and conclusions. We tried to emphasize the main achievements only.

While the introduction is engaging, it lacks clarity. The authors should aim for conciseness by reducing the length of some paragraphs. Additionally, the research problem and hypothesis need to be articulated more clearly. Please improve the readability of this section.

Answer:

The introduction was significantly revised. The main hypothesis was possible alternative way of intracellular delivery without fast biodegradation in cellular lysosomes with subsequent antigen presentation, innate immunity induction and trained immune response development. The endosomal-lysosomal entrapment during endocytic pathway remains significant problem for therapeutic drug delivery and vaccinology with possible risks of allergic and autoimmune disorders. One from possible solutions is membrane fusion between extracellular vesicles and plasma membranes of cells.

Regarding references, please clarify the phrase “[3 and references therein].”

Answer:

There are a lot of related references in the mini-review [3]. The article of Morozova O.V., Golubinskaya P.A., Obraztsova E.A., Eremeev A.V., Klinov D.V.  Structures, stability and cellular uptake of protein nanoparticles (NP) and extracellular vesicles (EVs). Current Drug Delivery 2024, 21 DOI: 10.2174/0115672018314957240508073903

is available on-line in open access. Therefore, it seems reasonable to include the only reference [3] instead of numerous papers because of limited volume of the research manuscript.

The drying conditions for the EV film appear to be critical. Can the authors confirm that the structures remain consistent with the initial evaluation? What potential impacts might there be on their physicochemical and biological properties?

Answer:

EV remain stable during isolation and storage under physiological iso-osmotic conditions whereas in deionized water hyposmotic shock results in EV destruction and leakage of their internal environment as shown by STEM, DLS and previously published [3]. The main problem during drying is EV concentrations. Higher concentrations may form multilayer films whereas diluted suspensions with lower concentrations do not create uniform monolayers of membranes. Porous layers with numerous gaps are not convenient for drug wrapping and packaging. Temperature and time of drying under reduced pressure appeared not to be critical. The sooner the better to keep conformations of transmembrane proteins. Exact EV concentrations are hardly possible to measure. Therefore, it seems reasonable to rely on lipid concentrations Phospholipids forming a complex with ammonium ferrothiocyanate were defined colorimetrically [9]. Their original concentration range 25- 125 nmole/ml is described in the revised manuscript. Protein concentrations varied in a broad range, differed for various small EV types and cell lines and, therefore, could not be considered to estimate dynamic range convenient for EV thin layer formation.

It would be beneficial for the authors to include a figure that outlines their experimental design to illustrate the steps of the study more effectively.

Answer:

General scheme of our research is shown in Graphical Abstract. It illustrates both isolation of EV from the conditioned culture media after growth of human embryonic and cancer cells by means of differential centrifugation at 20,800 g, their fluorescent labelling and subsequent cellular uptake using original cells (autologous combination) and foreign cells (heterologous pairs) preliminary stained with the lysosome-specific red fluorescent dye LysoRed. Interferon lambda (IFNÆ›) RNA transcription and protein production in the presence of heterologous EV isolated from foreign cells are depicted by oppositely directed arrows.

Were any steps taken to remove protein aggregates or non-EV contaminants from the centrifuged samples?

Answer:

Additional washes 3 times with cold PBS with subsequent sedimentation at 20,800 g at +4°C during 30 min are described in the section “Materials and Methods” of the revised manuscript.

How was the gradual accumulation of EVs quantified? Were fluorescence intensity values normalized?

Answer:

Fluorescent EV inside cells were quantified by spectrofluorimetry as described in the corresponding section “Fluorescent labeling and quantitation” of the “Methods”. To do so, the adherent monolayers of cells were washed with cold PBS 3 times to remove surface fluorescent EV and 200 ml of deionized water were added to each well of the 24-well culture plate. The excitation at 640 nm and maximum emission at 670 nm appeared to be optimal for Cy5-labelled EV. The fluorescence intensity values were not normalized since control human embryonic (Figure 4) and cancer (Figure 5) cells were not fluorescent at excitation at 640 nm and emission range 650-750 nm used for Cy5 detection. Figure 3 demonstrates original data of spectrofluorimetry as relative fluorescence units (RFU) shown on axis Y.

Were appropriate positive controls (e.g., LPS, Poly I:C) used to validate the interferon gene expression system?

Answer:

Interferon (IFN) gene expression was analyzed at both RNA transcription stage by reverse transcription with subsequent quantitative real time PCR and at protein translation stage using ELISA. Total RNA were isolated from control and experimental cells using guanidine isothiocyanate (GITC) lysis to dissociate nucleic acids from proteins and subsequent alcohol precipitation with washes with ethanol and aceton. Total RNA have typical UV spectra with maximum at 260 nm and are suitable for reverse transcription with MMLV reverse transcriptase. After the reverse transcription DNA copies were used in real time PCR with primer pairs and fluorescent hydrolysis probes specific for human IFN α, β, g and λ mRNA as previously published. LPS and poly I:C cannot be considered as positive controls for highly specific real time PCR. Human IFN protein production and secretion were evaluated in culture media of the control and experimental cells after addition of EV by ELISA with commercially available kits as described in details in our revised manuscript in the corresponding section “Enzyme-linked immunosorbent assay (ELISA)”.

No statistical analysis was provided. It is essential to address this to better demonstrate the effects of size and the significance of the findings.

Answer:

Continuous variables were compared using Student’s t-test. For categorical variables, the Chi square test with the Yates correction was used. P values <0.05 were assumed to be significant.

Please revise figure captions to correct typographical errors and address any misuse of abbreviations.

Answer:

Figure captions have been corrected.

For clarity, rather than referring to “lower/upper panel,” identify images as A, B, C.

Answer:

Done.

Figures 4 and 5 currently have suboptimal resolution; please revise accordingly.

Answer:

          Confocal fluorescent microscopy with magnification *600 and high quality immersion oil resulted in clear and evident patterns of the fluorescent EV intracellular distribution in the human adherent embryonic and cancer cells. LysoRed staining of lysosomes is commonly shown in red color whereas cyanine Cy5-labelled materials – in yellow color. Probably, mixture of the red and yellow colors in merged images on both figures 4 and 5 were not optimal. “Image J” software permits to change colors on images but alternative colors are not widely used in scientific illustrations of the fluorescent microscopy. Previous sizes beyond margins and localization of the figures were not satisfactory and were changed but common color codes for two fluorescent dyes remained the same.

          Other possible problem is lysosomes diameter range from 0.1 to 1.2 mm that is less optical microscopy limit. Fluorescent microscopy permits to see a part of labelled EV whereas others vesicles of smaller sizes may look like blurred fuzzy background.

The methodology for evaluating innate immunity activation was inadequately described, leading to confusion in the results. This section should be revisited.

Answer:

Interferon system including all 3 known types of human interferons IFN α, b, g and l were evaluated at levels of RNA transcription and protein translation. The methods of reverse transcription with subsequent real time PCR with fluorescent hydrolysis probes and ELISA with commercially available kits have been revised.

The discussion lacks depth and largely repeats the findings without adequately linking them to relevant literature. A more comprehensive analysis is needed.

Answer:

We tried to do our best to link our findings to relevant literature and to avoid unnecessary repeats from the text of the revised section “Discussion”.

The assertion that EVs evade immune detection warrants a more cautious interpretation until comprehensive immune assays, such as cytokine profiling and assessments of antigen presentation corroborate it.

Answer:

          The fusion of membranes of extracellular vesicles with plasma membranes of cells permitted to escape from endocytotic pathway as shown by different intracellular distribution of lysomomes and EV (Figures 4 and 5 – fluorescent confocal microscopy images). Lysosome entrapment and biodegradation are necessary for antigen presentation which is required for both innate and adaptive specific immune response induction. To our knowledge, at present there are more than 300 known cytokines. Therefore, comprehensive profiling of only innate immunity within one limited research is hardly possible. Besides specific T- cellular and humoral immune response together with trained immunity should be studied. Because of limited funding and volume of manuscript we focused on interferon system including all 3 types of human interferons IFNα, b, g and l at levels of RNA transcription (by using reverse transcription with subsequent real time PCR with fluorescent hydrolysis probes) and protein translation (by means of ELISA with commercial kits).

Authors are invited to include a brief discussion regarding potential limitations of their findings in light of proper data interpretation.

Answer:

Undoubtedly, further research is necessary. The revised discussion includes comprehensive comparison of new data from the research manuscript with already published papers. All suggestions and hypothesis seem to be careful with corresponding references and modal verbs.

Reviewer 2 Report

Comments and Suggestions for Authors

The present article, “Extracellular vesicles capable to escape from endosomal-lysosomal entrapment and innate immune response,” is interesting; however, the authors should address the following concerns

  • Line 25: The Authors suggested describing autologous and heterologous EVs
  • Line 168: Human interferons IFN α, β, γ, λ primer sequences should be provided. Western blotting will provide stronger evidence.
  • What are the n numbers for all the experiments? Include n numbers in figure legends.
  • Line 204: How frequently is the stability of exosomes tested? How did they test? Authors should provide details of test results
  • Representative images, along with bar graphs for Fig. 3, will give a better idea
  • Fig 4&5: Exos ICC fluorescence quantification in bar graph should be provided
  • Line 396: “The interferon IFNλ gene expression was found after addition of heterologous EV only”. Authors should explain clearly which experiments are with heterologous Exos and which ones are with autologous. It's confusing. 

Author Response

The present article, “Extracellular vesicles capable to escape from endosomal-lysosomal entrapment and innate immune response,” is interesting; however, the authors should address the following concerns.

Answer:

Thank you so much for your valuable suggestions for revision of our manuscript. We addressed every identified problem in the revised manuscript. These corrections can be seen in Microsoft Word Track changes for your convenience.

  • Line 25: The Authors suggested describing autologous and heterologous EVs

Answer:

Extracellular vesicles were isolated from the conditioned culture media after growth of human normal embryonal and cancer cells by means of differential centrifugation at 21,000 g. Then the vesicles were labelled with the fluorescent cyanine dye Cy5. To study cellular uptake and intracellular distribution patterns the fluorescent extracellular vesicles were added to culture media of original cells (embryonal vesicles for embryonal cells and cancer vesicles for cancer cells, so called “autologous combination”) and foreign cells (embryonal vesicles in cancer cells and vice versa in “heterologous” pairs). General scheme of our research is shown in Graphycal Abstract.

  • Line 168: Human interferons IFN α, β, γ, λ primer sequences should be provided. Western blotting will provide stronger evidence.

Answer:

Nucleotide sequences of both primer pairs and fluorescent hydrolysis probes specific for human IFN α, β, γ and λ had been previously published several times and are included in the revised manuscript.

  • What are the n numbers for all the experiments? Include n numbers in figure legends.

Answer:

At least, 3 repeats of each experiment were performed. The numbers are included in the corresponding figure legends.

  • Line 204: How frequently is the stability of exosomes tested? How did they test? Authors should provide details of test results

Answer:

Stability of extracellular vesicles was analyzed immediately after isolation by means of differential centrifugation at 20,800 g for 30 min, after fluorescent labelling in solution with pH 9.3 and during storage in PBS at +4°C weekly during the first month and later in each month by scanning transmission electron microscopy (STEM) and dynamic light scattering (DLS).

  • Representative images, along with bar graphs for Fig. 3, will give a better idea

Answer:

The figures have been revised accordingly.

  • Fig 4&5: Exos ICC fluorescence quantification in bar graph should be provided

Answer:

  To reveal intracellular distribution of extracellular vesicles and lysosomes ICC (immunocytochemistry) fluorescence with antibodies specific to certain transmembrane proteins or glycoproteins of the small extracellular vesicles was not used since simple low-speed differential centrifugation at 20,800 g using bench Eppendorf centrifuge resulted in complex mixtures of vesicles that may differ among various cell types. LysoRed fluorescent dye is pH-sensitive. Therefore, it becomes fluorescent at acid pH 4.5-5.0 but not in cytoplasm and other subcellular compartments with pH higher 7. Extracellular vesicles were labelled with the fluorescent cyanine dye Cy5 without antibodies as described in the section “Materials and Methods” of our revised manuscript.

  • Line 396: “The interferon IFNλ gene expression was found after addition of heterologous EV only”. Authors should explain clearly which experiments are with heterologous Exos and which ones are with autologous. It's confusing. 

Answer:

The same experiments were carried out with autologous and heterologous combinations of extracellular vesicles and host cells. As to innate immunity induction both reverse transcription with subsequent quantitative real time PCR with fluorescent hydrolysis probes and ELISA with commercially available kits were perform for different pairs to reveal possible enhanced delivery of the extracellular vesicle in their original host cells. However, this hypothesis was not confirmed.

Reviewer 3 Report

Comments and Suggestions for Authors

The authors isolated extracellular vesicles (EVs) using a two-stage differential centrifugation method on conditioned culture media obtained from the growth of adherent human embryonic and cancer cells. This process was conducted with a benchtop microcentrifuge, eliminating the need for high-speed ultracentrifugation. Techniques such as Scanning Transmission Electron Microscopy (STEM) and Dynamic Light Scattering (DLS) confirmed that the isolated EVs are classified as small EVs, with sizes ranging from 10 to 200 nm.

The EVs remained stable during storage in phosphate-buffered saline (PBS) at 4 °C for up to five months. However, exposure to deionized water, freeze-thaw cycles, and sonication led to the disruption of the EV's cell membrane envelopes. The authors suggest that the membranes could be dried in thin layers under reduced pressure and subsequently used for loading via the "thin film hydration" technique.

The authors reported that the EVs were not toxic to the human cell lines studied. Over seven days, the EVs gradually accumulated within the attached cells in monolayers, without reaching peak values. Confocal fluorescent microscopy analysis indicated that the intracellular distribution of fluorescent EVs did not coincide with lysosomes. Furthermore, the EVs' escape from lysosomal entrapment and biodegradation did not trigger antigen presentation or an innate immune response.

Strength of the manuscript: the authors applied a new microcentrifugation protocol, the test gives insight into the stability of Evs and into possible clinical application.

The weakness of the manuscript: the conclusion does not emphasize what is new compared to what has already been published about Evs.

Suggested corrections:

  1. Introduction: The authors could use a figure to show the different sizes of Evs or show the isolation methods in a table with the advantages and disadvantages of individual methods.
  2. The authors could schematically present the part of the discussion related to the entry of EVs into cells.
  3. Do the authors have any knowledge of how the aqueous solution of surfactants (below and above the critical micellar concentration) affects the stability of EVs, whether the membrane of larger EVs is more diffuse, so surfactants more easily penetrate the phospholipid bilayer and destroy the membrane, i.e. would larger EVs be more unstable than smaller EVs.

Author Response

The authors isolated extracellular vesicles (EVs) using a two-stage differential centrifugation method on conditioned culture media obtained from the growth of adherent human embryonic and cancer cells. This process was conducted with a benchtop microcentrifuge, eliminating the need for high-speed ultracentrifugation. Techniques such as Scanning Transmission Electron Microscopy (STEM) and Dynamic Light Scattering (DLS) confirmed that the isolated EVs are classified as small EVs, with sizes ranging from 10 to 200 nm.

The EVs remained stable during storage in phosphate-buffered saline (PBS) at 4 °C for up to five months. However, exposure to deionized water, freeze-thaw cycles, and sonication led to the disruption of the EV's cell membrane envelopes. The authors suggest that the membranes could be dried in thin layers under reduced pressure and subsequently used for loading via the "thin film hydration" technique.

Answer:

“Thin film hydration” is not our suggestion. The approach is discussed in the published articles [1, 4, 7]. Figure 2 of our revised manuscript shows scanning transmission electron microscopy images of bioinspired vesicles consisting of protein (bovine serum albumin and human immunoglobulin IgG) and gold nanoparticles covered with cellular membranes from extracellular vesicles thus providing direct evidences of wrapping and packaging of organic and inorganic nanomaterials within membrane vesicles.

The authors reported that the EVs were not toxic to the human cell lines studied. Over seven days, the EVs gradually accumulated within the attached cells in monolayers, without reaching peak values. Confocal fluorescent microscopy analysis indicated that the intracellular distribution of fluorescent EVs did not coincide with lysosomes. Furthermore, the EVs' escape from lysosomal entrapment and biodegradation did not trigger antigen presentation or an innate immune response.

Strength of the manuscript: the authors applied a new microcentrifugation protocol, the test gives insight into the stability of Evs and into possible clinical application.

Answer:

Thank you for appreciating our work. Hopefully, the fast and simple centrifugation using bench microcentrifuge might be useful for further research and implementation.

The weakness of the manuscript: the conclusion does not emphasize what is new compared to what has already been published about Evs.

Answer:

          The conclusion was revised to point out the main achievements of our research.

 Suggested corrections:

  1. Introduction: The authors could use a figure to show the different sizes of Evs or show the isolation methods in a table with the advantages and disadvantages of individual methods.

Answer:

Our manuscript was aimed at research with simple and fast new isolation method but not at review of already available numerous methods including differential ultra-high speed centrifugation, sucrose gradient density centrifugation, immunoaffinity capture, ultrafiltration, size exclusion chromatography, polyethylene glycol (PEG) co-precipitation and others. The comparison of various isolation methods had been already published and were cited in the corresponding references [1,7]. The isolated extracellular vesicles varied in the range 10-200 nm and belonged to small extracellular vesicles.

  1. The authors could schematically present the part of the discussion related to the entry of EVs into cells.

Answer:

General scheme of our research is shown in Graphical Abstract. It illustrates both isolation of EV from the conditioned culture media after growth of human embryonic and cancer cells by means of differential centrifugation at 20,800 g, their fluorescent labelling and subsequent cellular uptake using original cells (autologous combination) and foreign cells (heterologous pairs) preliminary stained with the lysosome-specific red fluorescent dye LysoRed. Interferon lambda (IFNÆ›) RNA transcription and protein production in the presence of heterologous EV isolated from foreign cells are depicted by oppositely directed arrows. The revised manuscript already contains 5 multi-panel figures and 2 tables.

  1. Do the authors have any knowledge of how the aqueous solution of surfactants (below and above the critical micellar concentration) affects the stability of EVs, whether the membrane of larger EVs is more diffuse, so surfactants more easily penetrate the phospholipid bilayer and destroy the membrane, i.e. would larger EVs be more unstable than smaller EVs.

Answer:

The described method of centrifugation under iso-osmotic conditions without surfactants allowed us to isolate the small extracellular vesicles without contamination with larger vesicles according to common definitions in the field. The research was aimed at intracellular delivery without endosomal-lysosomal entrapment and immunity induction. Therefore, membrane destruction with surfactants was not studied. Nevertheless, the suggested investigation seems to be important for further possible implementation.

Round 2

Reviewer 1 Report

Comments and Suggestions for Authors

The authors addressed almost all my comments. I have no further suggestions.

Author Response

Thank you for your time and attention. Your careful detailed analysis helped us to improve our revised manuscript.

Reviewer 2 Report

Comments and Suggestions for Authors

Authors responded to all the comments

Author Response

Thank you so much for your valuable suggestions for revision of our manuscript.